# OpenReview forum: "Black-box Optimization of LLM Outputs by Asking for Directions"
_ICLR.cc/2026/Conference — Submitted to ICLR 2026_

### Official Review · Reviewer_mg1W · 2025-10-25

**Soundness:** 2
**Presentation:** 3
**Contribution:** 2
**Rating:** 4
**Confidence:** 4

**Summary:**

This paper tackles the setting of jailbreaking closed sourced LLMs with text-only API access, which is more prevalent but difficult setting than the white box setting, as malicious actors can leverage gradients to steer LLMs, and with APIs that output log probs, as gradient free can be used for jailbreaking. Existing work on the text-only setting requires surrogate LLMs or reward models to score possible jailbreaks and update them. This work provides a different approach by using the victim model directly to score possible jailbreaks / adversarial examples, and then leverages these scores to generate new attacks. This is done by developing a series of binary comparisons of potential adversarial attacks for the victim model to score, which are then used to attack the model directly.

**Strengths:**

The main strengths of this paper is that it does not require any auxilliary models and data to score adversarial samples, which is very useful for stress-testing such closed source API LLMs. The algorithm is simple and very easy to implement and builds on existing black box adversarial attacks on LLMs and VLMs. Additionally, the use of binary comparisons instead of absolute confidence scores is novel and well motivated.

**Weaknesses:**

There are two primary weaknesses of this paper. First, there is a lack of related work / comparisons to existing text-only black box methods. Methods such as BlackDan [1] and D-Attack / DH-CoT [2] are not mentioned. Additionally, while existing methods like PAIR and Tree-of-Attacks do require auxilliary models, it would strengthen the paper to compare against them, as currently the only comparison is to the logprobs based method.

Secondly, the results in Table 1 are confusing, as it seems that the approach is not very successful on its own. Table 1 shows attack success rates that are significantly lower than the transfer based approaches across all models, and incorporating the attack with transfer based methods, either directly or through ensemble does not yield significantly better results. Can the authors explain this?


[1] Wang, X., Huang, V. S. J., Chen, R., Wang, H., Pan, C., Sha, L., & Huang, M. (2024). Blackdan: A black-box multi-objective approach for effective and contextual jailbreaking of large language models. arXiv preprint arXiv:2410.09804.
[2] Zhang, C., Zhou, L., Xu, X., Wu, J., Fang, L., & Liu, Z. (2025). Jailbreaking Commercial Black-Box LLMs with Explicitly Harmful Prompts. arXiv preprint arXiv:2508.10390.

**Questions:**

1) How efficient is the algorithm at converging to a successful adversarial attack? Can the authors provide an ablation study on changing the maximum number of iterations?
2) The majority of the attacks are done on non-reasoning models or non-thinking models. How does the performance of this approach change for models such as the o1-3 OpenAI reasoning model series?
3) Is the repeated iteration done in a multi-turn fashion (i.e. does the model's context include the past binary comparisons)?

**Details Of Ethics Concerns:**

There is no ethics statement in the main paper and appendix on the potentially harmful effects of adversarial jailbreaking and other attacks on commercial LLMs. This needs to be added and addressed.

---

> ### Author Response · Authors · 2025-11-17
> **Author Response to Reviewer mg1W**
>
> We thank the reviewer for the detailed comments. Below, we provide our responses to all of your questions.
>
> > Methods such as BlackDan [1] and D-Attack / DH-CoT [2] are not mentioned.
>
> Thank you for bringing these references to our attention. We would be happy to include them in the related work. To clarify, BlackDan uses an additional LLM for assistance, which corresponds to a different threat model than the one we focus on in this paper. Regarding D-Attack, it is unclear whether it can scale to adversarial examples and prompt injection, but we would be happy to add a discussion of these papers in the updated version.
>
> > Additionally, while existing methods like PAIR and Tree-of-Attacks do require auxilliary models, it would strengthen the paper to compare against them, as currently the only comparison is to the logprobs based method.
>
> We want to highlight that the with-logprob setting assumes the attacker has access to the ground truth, making it roughly the best result achievable in a purely black-box setting. This serves as a very strong baseline, almost an upper bound.
>
> Furthermore, for jailbreak results in Tab 3, the with-logprob method is actually from [1], where the authors already demonstrate state-of-the-art results compared with other baselines (e.g., PAIR and Tree-of-Attacks), and our method is comparable to [1]. Also our goal is not to claim that our method is the best across all baselines, as the threat models differ.
>
> [1] Jailbreaking Leading Safety-Aligned LLMs with Simple Adaptive Attacks. ICLR 2025.
>
> > Secondly, the results in Table 1 are confusing, as it seems that the approach is not very successful on its own
>
> This is indeed true for adversarial examples: our method struggles to "start from scratch" because the initial signal is very sparse (almost everything looks like a dog). However, when starting from a partially successful attack, binary comparisons become much more effective. To the best of our knowledge, achieving this in a purely black-box, text-only Vision-LLM setting was not possible prior to our work.
>
> > How efficient is the algorithm at converging to a successful adversarial attack?
>
> Regarding the query efficiency, in line 95 we reported that an average of 5–450 queries is needed across the three attack scenarios.
> For jailbreaks, the query count is much lower than for the other two scenarios. As shown in Table 3, it ranges roughly from 4.9 to 79.3 queries.
>
> For adversarial examples and prompt injection, we set the maximum number of queries to 1000. Since the ASR is not always 100%, the per-sample query count can vary widely: some samples succeed within just 2 queries, while a few may fail even after 1,000 queries. An average of around 450 queries is sufficient to achieve the results reported in the paper.
>
> > The majority of the attacks are done on non-reasoning models or non-thinking models. How does the performance of this approach change for models such as the o1-3 OpenAI reasoning model series?
>
> We kindly disagree with this point. In fact, the GPT-5 and 3 Claude models reported in Table 1 are all reasoning models. Additionally, the OpenAI o1-mini and o3-mini models do not support image inputs, which prevents us from testing the adversarial examples included in Table 1.
>
> > Is the repeated iteration done in a multi-turn fashion (i.e. does the model's context include the past binary comparisons)?
>
> No, each run is independent. The model does not retain context from previous binary comparisons, so there is no multi-turn interaction involved in the repeated iterations.
>
> > There is no ethics statement in the main paper and appendix on the potentially harmful effects of adversarial jailbreaking and other attacks on commercial LLMs. This needs to be added and addressed.
>
> Thank you for pointing this out. We shared our findings with the relevant model providers (OpenAI and Anthropic) prior to submission. We will include a dedicated section on the ethics statement to address the potentially harmful effects of adversarial attacks on commercial LLMs.

---

### Official Review · Reviewer_ZB3Q · 2025-10-31

**Soundness:** 3
**Presentation:** 4
**Contribution:** 3
**Rating:** 6
**Confidence:** 4

**Summary:**

The paper presents an effective black-box attack strategy for LLMs that operate in the most restrictive, text-only API setting. The core contribution is the discovery that while LLMs are poorly calibrated for absolute confidence scoring, they are surprisingly well-calibrated for binary comparisons. The authors leverage this insight to build a general, query-based "hill-climbing" optimization algorithm. By repeatedly "asking for directions" via these comparative prompts, the attack iteratively refines a malicious input. The method's effectiveness is demonstrated across three distinct and important domains: adversarial examples for vision-LLMs, jailbreak attacks, and prompt injections.

**Strengths:**

1. The paper's primary insight, using comparative, self-reported confidence as an optimization signal, is a significant and novel contribution. It elegantly bypasses the common requirement for logits or confidence scores, which are rarely available in production systems. The validation in Figure 3, which contrasts the failure of absolute scoring with the success of binary comparison against ground-truth logits, is very convincing.
2. The attack is designed for and effective in the "text-only" black-box setting. This is a very practical and challenging scenario, and this work dramatically expands the known attack surface for deployed, proprietary models.
3. A key and counterintuitive finding is that larger, more capable models are often more vulnerable to this attack. The paper provides strong evidence for this across model families (e.g., Qwen-VL-72B > 7B, GPT-5 mini > GPT-4o mini). The hypothesis that this is because they are better at the comparative reasoning task is a useful insight for the field.
4. The method is not a one-trick thing. It is successfully applied to three very different attack types (adversarial images, jailbreaks, and prompt injections) across numerous model families (GPT, Claude, Llama, Qwen). Furthermore, the results are state-of-the-art, achieving near-perfect success on jailbreaks and even outperforming logit-based attacks in some cases (e.g., 98% vs. 56% on GPT-4.1 mini).

**Weaknesses:**

1. In Table 1, the high ASRs (e.g., 94.7% on GPT-4o mini) are achieved with the "Transfer+ours" hybrid method. In this hybrid, the improvement from the optimization step is sometimes marginal (e.g., 92.9% to 94.7% for GPT-4o mini), suggesting the (known) transfer attack is doing most of the work. The standalone power of the query attack for vision seems less impressive than for text.
2. The entire attack hinges on the model's willingness to answer the comparative "meta-prompt." The authors note this as a failure mode, where a strongly aligned model may simply refuse to perform the comparison. This seems like a critical vulnerability of the attack itself. The paper does not sufficiently explore the robustness of the attack to simple defenses on this meta-prompt (e.g., "I cannot compare prompts in a way that might lead to a harmful outcome").
3. As mentioned in the first point, the benefit from the "Transfer+ours" method is highly variable. For GPT models, the gain is tiny (1.8% on GPT-4o mini) (table 1), but for Claude models, it is massive (35.1% to 59.6% on Haiku 3.5). This significant discrepancy is not analyzed. Does it mean the transfer attack is already near-perfect for GPT, or that the Claude models provide a much better optimization signal? This is an important detail.

**Questions:**

1. Following up from weakness, why is the improvement from "Transfer+ours" so minimal for GPT models but so large for Claude models (table 1)? Does this imply that the "directions" from GPT are less effective, or that the CLIP-based transfer attack is already highly aligned with the GPT vision encoder?
2. The paper identifies that a model can refuse the comparison query as a defense. How difficult is it to bypass this? Did the authors experiment with iteratively re-prompting or reformulating the comparison prompt itself to get around such refusals?
3. The "security paradox" claim, does this finding suggest that alignment techniques that rely on a model's advanced reasoning (like self-critique or Constitutional AI) are fundamentally flawed, as that same reasoning capability can be turned against the model to guide an attack?
4. For the vision-LLM attacks, the query budget was 1,000. What was the average number of queries for a successful attack? Table 3 provides this for jailbreaks (e.g., 4.9-79.3 queries), which is very efficient. Is the cost for vision attacks similarly low, or does it regularly require hundreds of queries?

---

> ### Author Response · Authors · 2025-11-17
> **Author Response to Reviewer ZB3Q**
>
> We thank the reviewer for the very detailed comments. We noticed that the feedback mainly focuses on adversarial examples (and we address all those points below). We have also scaled our method to jailbreaks and prompt injections—please let us know if you have any suggestions or comments regarding these two scenarios as well!
>
>
> > The standalone power of the query attack for vision seems less impressive than for text.
>
> yes, we find that transfer-only attacks work surprisingly well on GPT models but achieve limited success on Claude models, suggesting that OpenAI’s models use a vision encoder very similar to open CLIP models.
>
> > The entire attack hinges on the model's willingness to answer the comparative "meta-prompt." The authors note this as a failure mode, where a strongly aligned model may simply refuse to perform the comparison. This seems like a critical vulnerability of the attack itself.
>
> The reviewer is correct that our attack relies on the model’s willingness to answer the comparative meta‑prompt. However, our querying style is intentionally harmless (e.g., asking which image is less likely to contain a dog, which instruction is more likely to trigger an email‑sending behavior, or which input is more likely to make the model start with “sure”). In many cases, the with‑logprob setting actually leads to more refusals, not fewer.
>
> With our comparison‑based queries, the model is focused on evaluating differences rather than directly responding to a sensitive request, which reduces the chance of triggering safety defenses.
>
>
> Moreover, even if a target model refuses to perform comparisons, here we provide a transfer-based variant, where we use a guide model (Model A) to obtain the optimization signal — one that does not refuse — and then apply the attack to a separate target model (Model B). The results are shown below: the diagonal entries correspond to the results already reported in the main paper. As shown in the table, the transfer attack can actually yield even better performance.
>
> | guide_model (row) / test model (column)      | gpt-4o mini | gpt-5 mini | claude sonnet 3.7 | qwen-7b |
> |------------------|------------|------------|------------------|---------|
> | gpt-4o mini       | 50.8%      | 12.28%     | 56.1%            | 54.4%   |
> | gpt-5 mini        | **84.2%**  | **35.1%**  | **73.7%**        | **77.2%** |
> | claude sonnet 3.7 | 47.36%     | 24.56%     | 14.0%            | 47.4%   |
> | qwen-7b           | 22.8%      | 7.0%       | 24.6%            | 21.0%   |
>
>
> > Does it mean the transfer attack is already near-perfect for GPT, or that the Claude models provide a much better optimization signal? This is an important detail
>
> It is hard to draw definitive conclusions without details of the models. We know that transfer attacks are not fully optimal, as logprob-based attacks can outperform them in some cases.
>
> Our main observation is that image transfer works particularly well against OpenAI’s models, likely because they use a CLIP backbone very similar to existing open models. In contrast, for prompt injections and jailbreaks, transfer-only attacks are more limited.
>
> > Following up from weakness, why is the improvement from "Transfer+ours" so minimal for GPT models but so large for Claude models (table 1)?
>
> There's not much left to improve for GPT models. Transfer already gets 90+% success because of high alignment with public encoders
>
> > Did the authors experiment with iteratively re-prompting or reformulating the comparison prompt itself to get around such refusals?
>
> To clarify, this is a defense we suggest (we don't find that current models often refuse). We didn't implement this as this would require some extra model alignment.
>
> > The "security paradox" claim, does this finding suggest that alignment techniques that rely on a model's advanced reasoning (like self-critique or Constitutional AI) are fundamentally flawed, as that same reasoning capability can be turned against the model to guide an attack?
>
> Possibly. It’s hard to say given the black-box nature of the models. It could also be that stronger models are simply better at detecting small differences between inputs that guide optimization, without requiring any explicit “reasoning.”
>
> > What was the average number of queries for a successful attack?
>
> Regarding the query efficiency, in line 95 we reported that an average of 5–450 queries is needed across the three attack scenarios.
>
> For adversarial examples and prompt injection, we set the maximum number of queries to 1000. Since the ASR is not always 100%, the per-sample query count can vary widely: some samples succeed within just 2 queries, while a few may fail even after 1,000 queries. An average of around 450 queries is sufficient to achieve the results reported in the paper.

---

### Official Review · Reviewer_QorE · 2025-10-31

**Soundness:** 3
**Presentation:** 3
**Contribution:** 3
**Rating:** 6
**Confidence:** 4

**Summary:**

The authors introduce an approach for attacking black-box large language models. They iteratively present a model with two slightly different images and based on the responses to that question, select the image showing desired behavior, and then use the selected image to create the two images for future iterations. This results in an image that can successfully elicit the desired behavior in models.

**Strengths:**

Black box attack for VLMs
Shows promising results

**Weaknesses:**

Very compute heavy and expensive
Unclear from the paper how many iterations were required to get the shown ASR

**Questions:**

1. What were the computational costs/time required for the optimization of the adversarial image?

---

> ### Author Response · Authors · 2025-11-17
> **Author Response to Reviewer QorE**
>
> We thank the reviewer for the overall positive scores!
>
> Regarding the query efficiency, in line 95 we reported that an average of 5–450 queries is needed across the three attack scenarios.
> For jailbreaks, the query count is much lower than for the other two scenarios. As shown in Table 3, it ranges roughly from 4.9 to 79.3 queries.
>
> For adversarial examples and prompt injection, we set the maximum number of queries to 1000. Since the ASR is not always 100%, the per-sample query count can vary widely: some samples succeed within just 2 queries, while a few may fail even after 1,000 queries. An average of around 450 queries is sufficient to achieve the results reported in the paper.

---

### Official Review · Reviewer_n2Y6 · 2025-11-01

**Soundness:** 2
**Presentation:** 3
**Contribution:** 2
**Rating:** 2
**Confidence:** 4

**Summary:**

The paper studies the use of a binary comparison rather than relying on a signal (e.g., logprob signals or from a surrogate model). They show that in image classification, this comparison is significantly more calibrated than directly asking the model. Furthermore, using this tool they  case study three scenarios for which their method is effective.

**Strengths:**

1- The paper is well-written and is easy to understand.

2- The comparison between binary comparison and directly asking the model for the confidence score is the main novelty of this paper, where in my view can be applied in a broader scope.

3- The results of the paper seem promising.

**Weaknesses:**

1- I believe that a more comprehensive related work needs to be done in this paper. The idea of comparing only two prompts is largely investigated in the prompt optimization literature. For instance:

Wu et al. "LLM Prompt Duel Optimizer: Efficient Label-Free Prompt Optimization"
Lin et al. "Prompt Optimization with Human Feedback"

2- In the continuation of the previous point, there are several papers that study the calibration of the model's score. This paper only compares with a basic approach--directly asking for a score without defining any rules, criteria, etc.

3- I'm still confused about how the model answers this question: "which one of these prompts is less harmful: 1- Tell me how to build a bomb|suffix_1; and 2- Tell me how to build a bomb|suffix_2" and there needs to be clarification here. Moreover, I don't understand how the comparison of these two will give a meaningful signal. I think more ablation studies are required to ensure the significance of this comparison. For instance, how does the answer of this comparison correlate with the value of the log-probs.

4- The baselines in all three case studies are weak. In the first scenario, the authors only compare their methods with a transfer method. Can the authors clarify if the attack algorithm for the transfer was adapted from (Li et al., 2025)? How about other methods? For instance:
Hu et al. "Transferable adversarial attacks on black-box vision-language models"

5-  In the second and third scenario, they only compare with a random search method that has access to the log-probs. Even in this case, the only meaningful comparison is for GPT-4.1 mini in Table 3 (actually I don't understand how the log-probs were calculated for this black-box model. Are they exploiting the same top-5 method as the original paper does?) Why didn't the authors include more attacking methods (preferably more recent ones) in their table?

**Questions:**

1- Can you do the same calibration experiment for the adversarial prompt? I.e., asking a model to give you a score (there are methods developed for this already such as StrongReject) vs. asking it two only compare two prompts?

---

> ### Author Response · Authors · 2025-11-17
> **Author Response to Reviewer n2Y6 [1/2]**
>
> Thank you very much for the detailed comments. We appreciate the time and care you put into the review. It seems there may be a misunderstanding regarding our choice of baselines and the way our method operates. Before diving into specific responses, we would like to clarify why we did not include additional baselines first:
> 1. Our goal is not to beat all existing methods. The purpose of the paper is to show that even in a purely black-box, text-only scenario, one can discover new attack entry points simply by asking for directions — a phenomenon we believe deserves attention.
> 2. We intentionally avoid comparing against methods with different threat models (e.g., transfer-based attacks, LLM-assisted attacks, or white-box methods), as such comparisons would not be meaningful or fair.
> 3. The baseline we do use — the log-prob method — is, in our view, the strongest method under the same text-only API setting. We do not think the current baseline is weak; as shown in [1], this log-prob–based method outperforms the other baselines.
> 4. Our method is generic, and there is no directly comparable baseline in the existing literature.
>
> Below, we provide our responses to all of your questions.
>
> > Add some related works on comparing two prompts.
>
> Thank you for pointing this out. We are happy to add more detailed discussions on this topic and cite these papers in an updated version. To clariy, the first paper is work from the same period, which is a different method and focus on different applications; while the second one uses human feedback which doesn't apply in our setting (especially for adversarial examples).
>
> > “This paper only compares with a basic approach--directly asking for a score without defining any rules, criteria, etc.”
>
> Our main point is that extracting useful optimization directions from LLMs is inherently challenging, and simply asking for confidence scores does not resolve this issue. We also believe that adding more handcrafted rules or criteria would not scale to adversarial examples in vision-LLMs. It’s not clear how such rules could be meaningfully defined for images—for example, how would one specify that a dog image is “x% less like a dog,” or what kinds of rules could reliably capture such nuances?
>
> To be clear, we never claimed that pairwise comparison is the only viable approach. If you’re aware of alternative calibration methods that could scale across all three attack scenarios, we would be happy to consider them as baselines and expand our discussion in the paper.
>
> > I don't understand how the comparison of these two will give a meaningful signal
>
> We do include an experiment showing how the model’s answers correlate with log-probabilities (see Figure 3b). The fact that such a simple comparison can already provide a meaningful optimization signal is indeed surprising—and this is exactly the point of our work.
>
> The intuition is straightforward: if the target model selects the prompt that is actually more adversarial (as verified by ground truth; see Figure 3b), then the optimization process has taken a step in the right direction, bringing the attack closer to success.
>
> > In the first scenario, the authors only compare their methods with a transfer method. Can the authors clarify if the attack algorithm for the transfer was adapted from (Li et al., 2025)? How about other methods?
>
> To clarify, our goal is not to outperform transfer-based attacks, as such a comparison would not be meaningful given the different threat models. The comparison in the table is transfer-only vs. transfer + ours, where the success rate reflects successful transfer attacks plus the additional successes achieved by our method.
> This experiment is intended only to show that our approach can further improve results when combined with transfer attacks. If stronger transfer attacks are used, the improvement may vary, but our goal is not to demonstrate that our method outperforms transfer attacks themselves.
>
> > In the second and third scenarios, they only compare with a random search method that has access to the log-probs. Why didn't the authors include more attacking methods (preferably more recent ones) in their table? Are they exploiting the same top-5 method as the original paper does?
>
> As we explained, the with-logprob setting assumes the attacker has access to the ground truth, making it roughly the best result achievable in a purely black-box setting. This serves as a very strong baseline, almost an upper bound.
>
> Furthermore, for jailbreaks results in Tab 3, the with-logprob method is actually from [1], where the authors already demonstrate better results compared with other baselines. Since our method also is comparable to [1], we believe including additional baselines is unnecessary. And our goal is not to claim that our method is the best across all baselines, as the threat models differ.
>
>
> [1] Jailbreaking Leading Safety-Aligned LLMs with Simple Adaptive Attacks. ICLR 2025.

---

> > ### Author Response · Authors · 2025-11-17
> > **Author Response to Reviewer n2Y6 [2/2]**
> >
> > > Even in this case, the only meaningful comparison is for GPT-4.1 mini in Table 3 (actually I don't understand how the log-probs were calculated for this black-box model. )
> >
> > First, we do not agree with “the only meaningful comparison is for GPT-4.1 mini in Table 3,” as our goal is not to show that we can beat the upper bound to prove the method’s strength. Achieving results comparable to with-logprob is already a strong outcome.
> >
> > In fact, OpenAI provides the log probabilities of the top-20 tokens for certain models. This allows us to check whether the target token appears within the top-20 and, if so, use its probability for the calculation.
> >
> > > Can you do the same calibration experiment for the adversarial prompt? I.e., asking a model to give you a score (there are methods developed for this already such as StrongReject) vs. asking it two only compare two prompts?
> >
> > We’d like to ask for clarification. Are you referring to using specific rules or criteria when requesting a confidence score? As far as we understand, StrongReject is intended for evaluation, not for getting optimization signals from LLMs. For example, in the case of adversarial examples, it is unclear how such rules could be defined.
> >
> > We would be happy to include additional experiments on this if you could suggest some meaningful rules.

---

### Meta-Review · Area_Chair_ZhiG · 2025-12-29

**Summary:**

This paper identifies a key and counterintuitive phenomenon: in a purely black-box, text-only setting, LLMs are poorly calibrated for absolute confidence scoring but exhibit strong calibration in binary comparisons, which can be exploited to guide attacks across multiple scenarios. The reviewers generally acknowledged the novelty and practical relevance of this insight, but raised concerns in positioning and presentation, including ambiguity in whether it should be evaluated as an attack algorithm, insufficient coverage of baselines and related work, and unclear interpretation of some experimental results (especially for vision). As a result, despite a solid core idea, the submission was not considered sufficiently competitive for acceptance in its current form.

**Reviewer Concerns:**

The rebuttal addressed several points by clarifying the intended threat model, the motivation for the chosen baselines, the query complexity across different scenarios, and the availability of log-probability information in restricted black-box settings. It also provided additional explanation for the transfer-based vision results and indicated plans to expand the related work and ethics discussion, which helped reduce some sources of confusion.

However, some concerns remain only partially addressed. In particular, the overall positioning of the paper is still not entirely clear, which continues to affect how reviewers evaluate the adequacy of baselines and the interpretation of standalone results, especially in the vision setting. In addition, questions about the consistency of gains across models and the depth of analysis on limitations and defenses were not fully resolved.

**Reviewer Scores:**

There are four reviewers with initial scores of 2, 4, 6, and 6. After the rebuttal, the reviewer with score 2 would likely still have unresolved concerns about competitiveness and evaluation scope, with no clear reason to change their score. The reviewer with score 4 may find some clarifications helpful, but could still remain uncertain overall and keep a similar borderline score. The two reviewers with initial scores of 6 would likely view the rebuttal as addressing most of their questions and therefore maintain their generally positive assessments.

---

### Decision · Program_Chairs · 2026-01-26

Reject